# Practical Judgment of Workload Based on Physical Activity, Work Conditions, and Worker’s Age in Construction Site

**DOI:** 10.3390/s20133786

**Published:** 2020-07-06

**Authors:** Nobuki Hashiguchi, Kota Kodama, Yeongjoo Lim, Chang Che, Shinichi Kuroishi, Yasuhiro Miyazaki, Taizo Kobayashi, Shigeo Kitahara, Kazuyoshi Tateyama

**Affiliations:** 1Graduate School of Technology Management, Ritsumeikan University, Osaka 567-8570, Japan; 2Center for Research and Education on Drug Discovery, Faculty of Pharmaceutical Sciences Hokkaido University, Sapporo 060-0817, Japan; 3Faculty of Business Administration, Ritsumeikan University, Osaka 567-8570, Japan; lim40@fc.ritsumei.ac.jp; 4Kumagai Gumi Co., Ltd., Tokyo 162-0856, Japan; cho.sha@ku.kumagaigumi.co.jp (C.C.); skuroish@ku.kumagaigumi.co.jp (S.K.); yasmiyaz@ku.kumagaigumi.co.jp (Y.M.); skitahar@ku.kumagaigumi.co.jp (S.K.); 5Department of Environmental Systems, College of Science and Engineering, Ritsumeikan University, Shiga 525-8577, Japan; kobat@fc.ritsumei.ac.jp; 6Department of Civil Engineering, College of Science and Engineering, Ritsumeikan University, Shiga 525-8577, Japan; tateyama@se.ritsumei.ac.jp

**Keywords:** workload estimation, heart rate reserve, wet bulb globe temperature, construction hazards, worker safety

## Abstract

It is important for construction companies to sustain a productive workforce without sacrificing its health and safety. This study aims to develop a practical judgement method to estimate the workload risk of individual construction workers. Based on studies, we developed a workload model comprising a hygrothermal environment, behavioral information, and the physical characteristics of workers). The construction workers’ heart rate and physical activity were measured using the data collected from a wearable device equipped with a biosensor and an acceleration sensor. This study is the first report to use worker physical activity, age, and the wet bulb globe temperature (WBGT) to determine a worker’s physical workload. The accuracy of this health risk judgment result was 89.2%, indicating that it is possible to easily judge the health risk of workers even in an environment where it is difficult to measure the subject in advance. The proposed model and its findings can aid in monitoring the health impacts of working conditions during construction activities, and thereby contribute toward determining workers’ health damage. However, the sampled construction workers are 12 workers, further studies in other working conditions are required to accumulate more evidence and assure the accuracy of the models.

## 1. Introduction

The construction industry serves as the base for several other industries in every country and contributes significantly to the national economy. In order to maintain the industry’s productivity, it is essential to ensure the safety and health of its workers [1,2]. The need for construction companies to maintain safe working conditions and environment arises in view of the fact that the construction industry is more labor-intensive than the other industries. Since every site is exposed to these dangers, it is crucial to implement integrated safety management measures at construction sites.

Characterized by poor working environments, such as poor scaffolding, aerial work platforms, high humidity at high temperatures, and a worksite adjacent to heavy construction equipment, construction sites often contribute toward increasing the physical workload of construction workers [3,4,5]. A high-temperature or highly humid work environment and long-term physical workload expose workers to chronic fatigue, injury, illness, and health risk, and thereby reduce a site’s productivity [6,7]. An increase in the number of older workers, that is, aging of the construction workforce, has also been reported to affect the productivity of the construction industry [8,9]. Concerning the fatality rate in this industry, the U.S. Bureau of Labor Statistics reported that, between 2003 and 2010, more than 43,000 people experienced fatal accidents at the construction site [10]. In the United States of America, 37.9 percent of these workers were reported to experience fatigue. A study shows that a lack of attention to workers’ safety, health, and productivity can lead to fatal outcomes [11]. In Japan, according to a survey by the Ministry of Health, Labor, and Welfare, in 2017, 323 workers died in work-related accidents in the Japanese construction industry. It accounts for more than 30 percent of all industrial deaths in Japan [6]. Given the high frequency of occupational deaths and accidents, the construction industry is considered one of the most dangerous industries [12,13].

Concerning the physical workload, it refers to the physical strain that workers undergo while discharging their duties. The literature has evaluated physical workload by measuring different work elements, such as posture (standing, sitting, or bending), activity type (walking, running, or manual material handling), body weight, and exercise capacity [14,15]. Physical workload includes individual factors (for example, age, training, and nutrition) and environmental factors (for example, temperature, humidity, and noise) [16,17]. Thus, physical load factors comprise workload and external and internal load factors [14,16]. However, it is difficult to take into account all the individual and environmental factors affecting the body. For example, the impact of manual labor on workers’ health differs among workers doing the same work under the same working conditions as well as among workers doing the same job under different working conditions [17,18]. Astrand et al. [16] confirmed that every individual has a different workload capacity. When monitoring the physiological response of workers, a measurement based on heart rate [18,19] may provide an alternative way of understanding the factors influencing an individual’s workload capacity. For example, this study uses heart rate to understand the impact of the physical load of a job as well as that of the aforementioned personal and environmental factors [20,21]. Since it is difficult to identify the important factors affecting workers’ health in every situation, an analysis based on workers’ heart rate can be useful for understanding their work capacity. By using the heart rate as an indicator, we can determine the methods that can be used to influence workers’ health in various working conditions. A method to quantify the physical work capacity of construction workers can provide an understanding of the means that can be employed to ensure the safety and health of the workers. In turn, this approach can help construction companies to maintain a healthy workforce while maintaining organizational productivity.

In this regard, it would be critical to use a sensing technology that can accurately record the heart rate of workers. Advancements in embedded biosensor systems (e.g., heart rate and skin temperature sensors) have led to the development of wearable devices that can measure the impact of physical workload [22,23]. These wearable health devices are commercially available at an affordable price in the form of lightweight wristbands. They provide work comfort when in contact with the skin [18]. In these smart watches, the sensors measure the acceleration resulting from the physical activity of the user, the electrical signals capture the electrocardiogram (ECG), and a photodiode and a light-emitting diode facilitate a photoplethysmography (PPG) probe (photoelectric volume pulse wave) [24]. Although the measurement accuracy and the resolution of these devices are inferior to general medical devices, they facilitate heart rate monitoring, an ECG, and the collection of various biological information such as arterial oxygen saturation (SpO2).

As an indicator, the heart rate reserve (%HRR) has been reported to be a major predictor for estimating an individual’s workload capacity [25,26]. This method assumes resting HR_resting_ (i.e., minimum HR during resting) as a level with no physical load, and calculates a percentage of the difference between working and resting HRs among HR reserve (i.e., HR reserve indicates the difference between HR_max_ and HR_resting_) [25,26,27]. Considering the health risks of workers, workers with 30–60% HRR should not be exposed to a higher physical load for a long period; similarly, workers with more than 40% HRR should not undertake any heavy workload exceeding 30 min [26,27]. In this study, workers with more than 40% HRR were considered to be at health risk. Although %HRR is a useful determination method, most studies often identify HRR based on certain activities such as walking, jogging, or treadmill. Given the frequent inflow and outflow of workers at construction sites, it often becomes difficult to use specific activities to measure the HR_max_ and HR_resting_ for each worker precisely. Hence, it is difficult to use %HRR as an indicator of workload at construction sites.

The study aims to develop a practical judgement method to estimate the workload risk of individual construction workers in real-time. It uses a model composed of workers’ movement acceleration, age, body mass index, and the wet bulb globe temperature (WBGT), which take into account temperature, humidity, and radiant heat. By using the model, the impact of workload risk is estimated through the worker’s %HRR. Studies have measured the movement acceleration and heart rate of the workers in order to investigate their efficiency by the layout of the workspace and the work-type [28]. However, few studies have reported a method to determine the impact of workload risk in typical construction site work environments [29,30]. In order to formulate strategies to manage workers and improve their productivity, it is important to determine the impact of workload on their health using simple and accurate methods.

## 2. Materials and Methods

### 2.1. Measurement System

Table 1 shows a list of measurement equipment and infrastructure considered in this study. In order to evaluate the type of work environment, the study measured the air temperature, relative humidity, and WBGT; HRR and acceleration (ACC) detected from the respiratory rate (R-R) interval in ECG were measured in order to determine the workload. In Table 1, the data acquisition device synchronized the time data to ensure that the data acquisition time did not differ between the workers.

We measured the heart rate and the physical activity of workers on the basis of the ECG signals captured using the smart clothing worn by construction workers, as shown in Figure 1A. The smart clothing is an underwear-type shirt integrated with biometric information sensor (detection of heart rate) [31,32,33] and a 3-axis acceleration sensor. Since the smart clothing is made of stretchable fabric, stretchable ECG electrodes were integrated with the hardware for measuring the heart rate [34]. The heart rate was detected by detecting the R-R intervals in the ECG signals. HRRs during load and rest were obtained by converting the detected R-R intervals. We attached a small heartbeat sensor device (WHS-2) to the smart clothing; we monitored the heart rate and the amount of physical activity by taking the 3-axis acceleration showing the spacing between the R-R waves of and the physical activity the subjects [35]. Using a Bluetooth low energy device, the heart rate and 3-axis acceleration data were sent to the data acquisition device used by the workers (CS2650, Texas Instruments, Inc., Dallas, TX, USA). Subsequently, data from the data acquisition device were transmitted to and stored on the server installed on network using the established wireless access point (data transfer device) in the work area. The measurement device and system configuration used in this study are shown in Figure 1B.

Based on the data provided by workers on their height, weight, and age, we calculated their the body mass index (BMI); the WBGT was calculated based on their labor time. In order to grasp the temperature and the relative humidity of the work environment, the WBGT was measured at 5-min intervals.

### 2.2. Measurement Method

Table 2 shows the measurement parameters and techniques used in this study. The participants were expected to be aware of their body measurements as they worked in construction companies that conducted these measurements on a regular basis. We asked questions related to age, height, and body weight of the subject and, subsequently, used the responses to determine the BMI. As per a study [36] on an East Asian cohort, people with a BMI between 22.6 and 27.5 had the lowest death risk, whereas people in a higher BMI range were at a higher risk of deaths from cancer, cardiovascular diseases, and other causes. The BMI was added to this study’s explanatory variables to examine worker health risk.

Heart rate during work (HR_working_) was determined by the average value of the heart rate measured every 5 min for each worker. Thus, the heart rate during work denotes the average of the data collected every 5 min; this data has been corrected by deleting null data or outliers [37,38]. An accurate measurement of the HR_max_ is not suitable for construction workers who are often in flux. Therefore, the HR_max_ was predicted using the equation of Tanaka [39]. In order to confirm the stability of the heart rate at rest time, the subject’s heart rate at rest time was measured more than thrice at 5-min intervals; the lowest average heart rate was used to determine the HR_resting_.

By using the 3-axis acceleration sensor for continuous monitoring, the physical activity level of the subject was evaluated to get ACC [40] of the three axes (the longitudinal axis: X, the lateral axis: Y, and the vertical axis: Z); the resulting ACC, which is physical activity, was calculated by an average value generated during the 5-min intervals. The intensity level of the physical activity is shown to be a predictor of good health [41]. The 3-axis acceleration method observed the strength of each worker’s overall movement during the working hours. Unlike office workers, construction workers can provide a more detailed operating data than those derived from the measured parameters; this data may include data on their capacity to lift load up the stairs. In other words, the method captures the intensity of the workers’ movement between the work activities (for example, walking, standing, and crouching) performed during the working hours.

### 2.3. Participants

The data were collected at the water injection pump construction sites of a construction company in the following dates of the year 2018: May 25th, June 29th, and November 16th (Kumagai Gumi Co., Ltd., Tokyo, Japan). The participants were tasked with the dismantling of the steel scaffolding. Specifically, eight steeplejack workers performed repetitive tasks (of those involved in the demolition) and four assistant workers carried out indirect work such as equipment installation. These 12 participants were notified of their selection as subjects in the experiment.

### 2.4. Protocol

Before data collection, we examined the potential risks to and discomfort of the subjects and the privacy issues from the data collection. It was confirmed that the smart clothing was a non-invasive device, and hence it did not hinder the progress of the construction work. In accordance with the Declaration of Helsinki, the human genome, and the Universal Declaration of Human Rights, data collection protocols such as heart rate, physical activity, and privacy conditions of the subjects were approved by the Ritsumeikan University research ethics examination number: BKC-2017-071-1. The approved protocol as well as the informed consent form distributed to all the subjects before data collection contained a description of the rights of the participants and ensured data confidentiality. Concerning privacy concerns, in order to minimize the risk of a disclosure of the personal information of workers, we did not use workers’ names; instead, a personal identification code was assigned to each subject (identifier) for data collection and analysis.

### 2.5. Data Collection

Table 3 shows the dates of data collection, ages of the subjects, work tasks, body measurements, duration of data collection, break time, work tasks, and the dataset each subject. The data were measured from 8:30 am to 5:00 pm, and the data were collected during the entire period or at any of the 5-min intervals in the time zone of half of the period of the working day. All the subjects were men; they were asked about their age, job title, and the provision of information about their body and weight. Since the cardiopulmonary function aims to eliminate the unhealthy subjects from the measurement, the subjects were also asked about the presence or absence of the history of cardiovascular disease and their current health condition (for example, whether they suffer from chronic cardiovascular disease). Among the 13 workers who expressed a desire to participate, 1 subject with arrhythmia did not participate in the experiment. Except for the preparation time for data collection, 5 min were given to each of the 882 datasets collected from 12 subjects.

When measuring the subject, we checked for the Hawthorne effect [42]. In this experiment, we did not monitor the activity of the subject; we waited a little farther from the work area and recorded and photographed the work with the help of two cameras installed in the work area. In general, manual laborers performing high-load work have their own health concerns; they also focus on the physical workload resulting from their daily operations. Hence, before starting the measurements, we instructed the subjects not to depart from the usual work patterns; they were also informed that the study did not intend to measure their productivity but their physical workload.

### 2.6. Model Development and Statistical Analysis

The independent variables comprised subjects’ ages, BMI, the amount of physical activity during labor, and the WBGT in the field environment. The binomial logistic regression was used to analyze the health risks determined by the level of %HRR. Logistic regression assessed the value of new medical treatments. It is one of the regression analysis methods evaluating the factors affecting a problem surrounded by controversies [43]. Logistic regression modeling is not limited to the study of physiological medicine, but it is also used in biology, engineering, ecology, health policy, linguistics, and business and finance [44]. Regression analysis has become an integral element of data analysis for describing the relationship between the independent variables and one or more of the explanatory variables.

A prediction model considering the parameters of these independent variables improves the accuracy of risk detection; in this study, we observe a strong correlation between risk factors. Therefore, it is necessary to consider the possibility of multicollinearity while conducting the statistical analysis. When developing the statistical model, the Pearson’s correlation coefficient and variance inflation factors (VIF) were calculated for determining the physical activity ACC, AGE, and BMI of the subjects and WBGT in the work environment.

The SPSS Version 26 for Windows (IBM), Excel add-in software “Excel Tokei 2015” (SSRI Co., Ltd., Tokyo, Japan), and modeFRONTIER^®^ (IDAJ Co., LTD, Yokohama, Japan) [45] were used as tools for conducting statistical analysis.

Table 4 shows the results obtained with respect to the correlation coefficient and the VIF between independent variables. A weak negative correlation coefficient (−0.333) was observed between physical activity ACC and the AGE of the subjects, while there was no correlation against the BMI and WBGT. The BMI of the subjects was slightly positively correlated (0.415) with the AGE and weakly correlated with the WBGT (−0.387). Furthermore, it was observed that the AGE of subjects was hardly correlated with the WBGT. The VIF serves as a reference for checking multicollinearity; the values of all the independent variables were the extent of the value 1–2, indicating a low likelihood of multicollinearity [46].

## 3. Results

### 3.1. Subject Characteristics and Measurements

Concerning S1–S8 scaffold workers and L1–L4 construction assistants, Table 5 shows the measurements associated with the heart rate, physical activity ACC, and the correlation coefficient r_ACC-%HRR_ between ACC and %HRR. The average and standard deviations of the data on the effect of the amount of physical activity on the heart rate were determined during the data collection spanning the entire working hour, including the break time.

Scaffold workers S1–S8 were compared with L1–L4 assistants, whose average and standard deviations between the ACC and the %HRR were large and their workload in the scaffold dismantling was found to be relatively high. The average% HRR of S5 and S7 was more than 40%; it implied that they were exposed to a high physical workload. Among the other workers, the %HRR of several workers exceeded 40% frequently; it implied that the workers were exposed to safety and health hazards. S2 was assigned to the foreman responsible for monitoring progress work in the construction site. On the day of the experiment, S2 participated twice in the meeting spanning for about 30 min, and S2 was not exposed to a heavy physical workload; therefore, S2′s heart rate and physical activity were not impacted.

While the amount of physical activity of S5–S8 was relatively small, their %HRR was slightly higher at 36–45%. On the day of data collection, they were engaged in manual labor for a long time at the scaffold tower. Some of the subjects were engaged in removing the steel scaffolding materials and placing those materials under the scaffold stairs; this work entailed a continuous exchange of steel materials among the subjects. Although this physical activity did not involve a lot of movement, it was laborious to dismantle the steel materials; hence, a high heart rate was recorded for these subjects.

#### 3.1.1. The Relationship between ACC and %HRR

Figure 2 shows the results of the %HRR for the ACC measured in each subject. The relationship between the ACC and the %HRR is shown in linear approximation; Table 5 describes the correlation coefficient r_L1–L4_ of L1–L4 and the correlation coefficient r_S1–S8_ of S1–S8. In all the subjects, the %HRR increased according to the increase in the ACC, and the correlation coefficient r of the ACC and the %HRR was at about 0.7–0.9. When compared to assistant workers, the higher heart trend of the scaffold workers shows their exposure to a high physical workload. During the break time, the workers’ ACC and %HRR were relatively low and their physical activity and heart rate at rest time were stable.

Figure 2 also shows that, despite an increase in ACC from S5 to S8, the %HRR tends to be relatively large. S1 recorded the highest ACC among all subjects because S1 not only placed the dismantled steel materials under the scaffold stairs but also carried them to the collection location. It is presumed that there was an increase in the transportation task between the unloading location of the scaffold stairs and the collection location, which led to an increase in both %HRR and ACC. S1 is the youngest among the subjects; as HR_max_ increased by the calculation based on age, there is a possibility that even same HR_working_ is no longer conspicuous when compared to the other subjects. Conversely, although relatively high-age assistant members, such as L3, had a relatively low ACC without a big movement behavior, S1′s %HRR was seen to be higher than the other assistant workers. This can be attributed to the fact that HR_max_ obtained for L1′s age is lower than that for the other subjects.

#### 3.1.2. The Relationship between ACC and %HRR by Workers’ Age

Concerning the younger and older age groups, Table 6 shows the number of subject’s data, the average and standard deviations regarding the age, ACC, %HRR, and the correlation r_ACC-%HRR_. The workers were divided into two age groups of 20–39 years and 40–59 years. In the younger age group (20–39 years old), %HRR is relatively low, while the amount of physical activity is the highest, when compared to the relationship between the ACC and the% HRR in the older age groups. Data on the ACC and the %HRR were significantly different between AGE_younger_ and AGE_older_.

For the two age groups, Figure 3 shows the relationship between the ACC and the %HRR in linear approximation and describes the correlation coefficient r of each subject of the two age groups. The correlation coefficient between the two age groups is separated by 20–39 years and 40–59 years of age; it is indicated by r_younger_ and r_older_. The correlation coefficient of 0.588 was seen to have a moderately strong correlation in the young age group; the correlation coefficient r of the relationship between the ACC and the %HRR increases with an increase in age, and the correlation tends to be stronger. The Figure 3 shows an increase in the change in %HRR against the ACC of old age group (40–59 years old), compared to the younger age groups. This relationship suggests that the %HRR increased with an increase in the workers’ physical activity and age.

#### 3.1.3. The Relationship between ACC and %HRR by Workers’ BMI

For each BMI group of workers, Table 7 shows the number of subject’s data, average, and standard deviations in regard to the ACC, the %HRR, and the correlation. The BMI groups are divided into three layers— ≤22.5 kg/m^2^, 22.6–27.5 kg/m^2^, and ≥27.6 kg/m^2^ [36]. Regardless of the BMI, a large difference was not found between the value of the %HRR and the ACC. The data on the ACC and the %HRR were non-significant among BMI groups, except for the relationship between BMI_middle_ and BMI_high_ in regard to ACC.

For each BMI group, Figure 4 shows the relationship between the ACC and the %HRR in linear approximation and describes the correlation coefficient r for each age group of subjects. The correlation coefficient of each BMI group is shown by r_low_, r_middle_, and r_high_, respectively. All correlation coefficients are between 0.6–0.8, approximately, in any BMI group; the relationship between the ACC and the %HRR showed a moderately strong correlation. Although the correlation coefficient r tended to be greater in the groups with higher BMI subjects, we did not see a significant change in %HRR for the ACC due to the BMI. This relationship suggests that the workers’ BMI is not affected when there is an increase in the %HRR due to an increase in the physical activity.

#### 3.1.4. The Relationship between ACC and %HRR by Workers’ WBGT

Table 8 shows the number of the subject’s data, the average, and standard deviations regarding the WBGT, the %HRR, the ACC, and the correlation coefficient r between %HRR and ACC. At the time of measurement, the time subjects spent on heavy work accounted for 75% of the total working hours, and the rest of the quarter was the rest time. Based on the limit of WBGT of ISO7243 (Indoor Environments) [47], the WBGT was divided into two groups—low and high—at 25.9 °C. The data on ACC and %HRR were significantly different between WBGT_low_ and WBGT_high_. Regardless of the BMI group, at high WBGT, a large difference was not significant for %HRR and ACC. Figure 5 shows the relationship between the ACC and the %HRR per WBGT group in linear approximation and describes the correlation coefficient r for each age group of the subjects. The correlation coefficient is shown by r_<25.9_ and r_≥25.9_. The correlation coefficient r became r_<25.9_ = 0.703 and r_≥25.9_ = 0.750, in any WBGT group; we see a strong correlation between the ACC and the %HRR. As WBGT became higher, the change in %HRR, which is larger than ACC, led to an increase in the heart rate. This relationship suggests that the %HRR increased with an increase in the workers’ physical activity and WBGT.

### 3.2. Logistic Regression Model

Logistic regression analysis was performed using the statistically significant (*p* < 0.05) independent variables in Table 4. The logistic model formula used in this study is shown below.
Workers’ health risk (≥ HRR 40%: 1, < HRR 40%: 0) = f(ACC, AGE, BMI, WBGT)(1)
where the objective variable is 1 for the worker’s health risk (≥ HRR 40%), and it is 0 for no health risk (< HRR 40%).

The independent variables are physical activity ACC, AGE of subjects, BMI calculated from height and weight, and WBGT in the work environment. Using the HR_max_ and HR_resting_ of each subject and calculating their %HRR, the independent variable was estimated by whether HRR ≥ 40%.

Table 9 shows the estimation results of the model on the health risks of workers examined in this study. In the Model 1, ACC, AGE, BMI, and WBGT were independent variables. By *p*-value, the ACC, AGE, and WBGT of the subjects of the work environment were statistically significant, but the BMI of workers was not significant. Wald χ^2^ shows the contribution of each variable to the model, and it indicates that the larger the value, the higher is its importance [48]. In the Model 1, compared to AGE and WBGT, the contribution of the ACC was relatively high. To determine the influence of the dependent variable, we use the odds ratio in the logistic regression. It is indicated that independent variable increases due to an increase in the odds when OR > 1; hence, the odds ratio can be a measure of a likelihood of a decline in independent variable with an increase in the odds when OR < 1 [43,44,49]. When the odds ratio is larger than 1, while the lower limit of the confidence interval (CI) is not less than 1, the ACC, AGE of workers, and WBGT in the work environment are independent variables. However, the BMI of workers is the independent variable whose odds ratio is less than 1—the lower limit does not exceed 1. By the results, without BMI, the Model 2 examined a model based on independent variables ACC, AGE, and WBGT. It is indicated the statistical significance of the three independent variables; AGE, ACC, and WBGT may be high to show the health risks of workers, which is important for the Model. Furthermore, according to the odds ratio, the influence of the WBGT on the health of workers was the largest.

Concerning the estimation results of the model, Table 10 shows the suitability index of the model due to its goodness of fit (GoF). The positive discrimination rate by the estimation of Model 1 was 88.9%; the positive discrimination rate by the estimation of Model 2, except the BMI that did not have statistically significant results, increased to 89.2%. Three indicators were determined to test the significance of the model by GoF. Akaike’s information criteria (AIC) = −2logL + 2k were defined, and AIC were considered as indicative of the model fit [50,51]. Here, k denotes the number of parameters in the model; the first term model represents the true goodness of AIC, and the second term represents the penalty due to an increase in the variable. The values of the AIC are small. According to the obtained AIC, the adaptation of Model 2 was slightly better than the Model 1. The Cox–Snell R^2^ corresponds to determining coefficients of a linear regression analysis R^2^, referred to as pseudo-R^2^. The fit of the model becomes better as the Cox–Snell R^2^ becomes larger; the Cox–Snell R^2^s of Model 1 and Model 2 are only 0.4–0.5, while the fit of the independent variable for the dependent variable was not very high [52,53]. However, when Nagelkerke R^2^ approaches from 0.5 to 1, the fit becomes higher [54,55,56]. The fit of the independent variable for the dependent variable of Model 1 is 0.590 and the one of Model 2 is 0.589. Both of them were high. Both the fit of Model 1 and Model 2 were good by these results.

## 4. Discussion

This study shows that a continuous measurement of the physical load of construction workers can change work conditions and increase an understanding of their health conditions. It measures the load fluctuation of the workers by using data collected from the smart clothing; this fluctuation corresponds to workers’ age, the temperature of the working environment, and working conditions (e.g., foreman and the real workers and the difference between the assistants and the scaffold workers). In line with the results of previous studies, this study shows that the physical demands differs in case of each worker; hence, results seen in previous studies on wearable devices (wristband-type devices) correspond [5,56,57,58] to this study.

Based on the work patterns (dismantling, transportation task, and the percentage of work activities, including preparation work), the physical demands of the workers vary even in the same construction site. In order to understand the health risks of workers, there is a need for continuously measuring a work in progress.

When measuring the subjects, we observed that the physical demands should not be sustained for a longtime. Concerning the workplace and physical demands, specific guidelines were created. (i.e., the work activities that increase the %HRR beyond 40% should be limited to 30–60 min [27].) Based on these guidelines, as shown in the subjects S5 and S7 (Table 5 and Figure 2), there were several workers with more than 40%HRR who continued to work throughout the day. Therefore, in order to reduce the high physical demands of these workers, some intervention must be implemented. This study provides insights on the appropriate interventions required for managing excessive physical requirements.

Most studies determine %HRR based on typical activity patterns and experimental environments, and there are very few cases that consider an actual construction site. Concerning the application of methods for determining %HRR of construction workers, it is difficult to measure HR_max_ and HR_resting_ [59,60,61] in advance for workers, and it is likely that %HRR will interfere with actual use as the indicators.

The value of this study’s important finding is that the use of covariates in logistic regression instead of %HRR rate has revealed hypotheses about the effects of physical workload on workers. The relationship among these covariates influence the heart rate of workers [62,63,64]. WBGT has the largest odds ratio among all the covariates and the impact on workers’ %HRR was significant [36]. To the best of our knowledge, this is the first study to report how the physical activity, workers’ age, and WBGT could be used to determine workers’ physical workload without calculating their %HRR. Concerning ≥40%HRR, the accuracy rate is about 89.2%, based on the estimation of the judgement model of the health risks of workers. Thus, in an environment where HR_max_ and HR_resting_ cannot be measured, it was indicated that the health risk could be judged by this study. The practical implication of the study is that instead of the heart rate reserve used in previous studies, we have very easily proposed a tool for measuring the health risk of construction workers.

This research is useful as an element of integrated management technology for the entire construction site using Information and Communication Technology (ICT) tools. For many applications in the construction industry, the introduction of construction robots, cooperation between big data and artificial intelligence, Machine to Machine (M2M) communication, and the introduction of Internet of Things (IoT) are expected to further advance the use of technology in the new ICT field. Building Information Modeling (BIM) and Civil Information Modeling (CIM) play a role in connecting the production process of construction, where research, planning, design, construction, and maintenance were separated, and are considered to contribute greatly to improving productivity [65]. The workers’ biometric data obtained in this research make it possible to visualize the health condition and workload of construction workers as a quantitative workforce, and to link them among construction sites. Based on the measurement data of field workers, construction companies can use it to simulate the construction process, predict the process progress, and estimate worker safety. By introducing ICT, by linking and visualizing the quantified workforce in the construction company, the convenience of HRM at the construction site is improved, and efficiency, labor saving, quality improvement of the construction industry, further contributes to improved safety. As technology advances, sensors are becoming smaller, high speed, and precision, and construction companies are increasingly required to have the evaluation skill to use the acquired data. In addition to investing in the use of sensing technology and investing system construction, human skill development that takes advantage of them is also an important issue.

We have several limitations in our study. First, the sampled Japanese construction company had only 12 workers in total. Due to this limited number of subjects, the dataset used the average of the physical activity and heart rate collected at 5-min intervals. We recognized that the observation period is sufficient for analyzing workers because previous studies have used data on heart rate and physical activity collected at about 30-min and 5-min intervals, respectively [25,26,27,38,66]. However, since the measured values are averaged, rapid changes in the worker’s condition could not be observed. Second, some physical activity and heart rate data were missing, which might have led to measurement errors. However, almost the same results were obtained even when these outliers were included in the analyses (data not shown in table). These outliers may slightly affect the heart rate mean or standard variance. To avoid these technical errors, there is a need to monitor more accurately the heart rate and physical activity. Third, the study used a self-reporting method that could result in differences between workers’ information on their height and weight. Future research should seek to include observable data to better understand the potential impacts of the physical workload on workers. Finally, this study provides an insight into the degree of contribution of the physical activity and other variables for estimating workload. Workers’ health is very fragile, and it is affected by their physical workload, mental state, and lifestyle. Hence, it is necessary to carry out further study in this regard. Further studies in other working conditions are required to accumulate more evidence and assure the accuracy of the models.

## 5. Conclusions

In this study, the heart rate and physical activity, the age, BMI, and WBGT of the working environment were measured for workers of a Japanese construction company. Given the high workforce mobility in the construction industry, this study developed a new judgement model of workers’ health risks as an alternative to %HRR. By using workers’ physical activity, age, BMI, and WBGT of the work environment as independent variables, it can be easily observed the physical load of worker without preparation such as HR_max_ and HR_resting_. It measured the heart rate and physical activity of construction workers by using a smart clothing equipped with biological and acceleration sensors. By logistic regression analysis, the risk to health by physical workload was analyzed. The results showed that physical activity, age and WBGT are important parameters of workload estimation. However, BMI of workers was not statistically significant, and hence it did not have a significant impact on the estimation of the health risks posed by the workload.

This study aimed to develop a method to facilitate a practical estimation of the workload risk of individual workers at a construction site in real-time. The use of a lightweight wearable device in this study has important theoretical implications in that it presents a real-time monitoring mechanism for examining workers’ health condition. This monitoring mechanism is very easy, without special preparation. It can also be adopted by firms to minimize the workers’ health damage.

In managing the health and safety of workers, it is useful to assess workers’ workload and health state quantitatively. Although there are several studies on workforce management [67,68,69], few studies focus on using workers’ individual heart rate, physical activity, and body measurements. Further research on the use of these attributes can improve the identification of the health risks of workers quantitatively and promote the productivity of workers at construction sites.

This study is the first study to report how to use physical activity, worker age, and WBGT to determine a worker’s physical workload without calculating %HRR. The results enable pragmatic decisions for real-time estimation of workload risk for individual construction workers. The value of this study is that it revealed that a quantitative workforce can visualize the health and workload of construction workers. The theoretical and practical implication are that instead of the heart rate reserve used in previous studies, we have very easily proposed a tool for measuring the health risk of construction workers. As a result, it becomes possible to connect and visualize the quantified workforce in the entire construction company, which leads to rationalization and safety improvement of the entire construction industry.

## Figures and Tables

**Figure 1 sensors-20-03786-f001:**
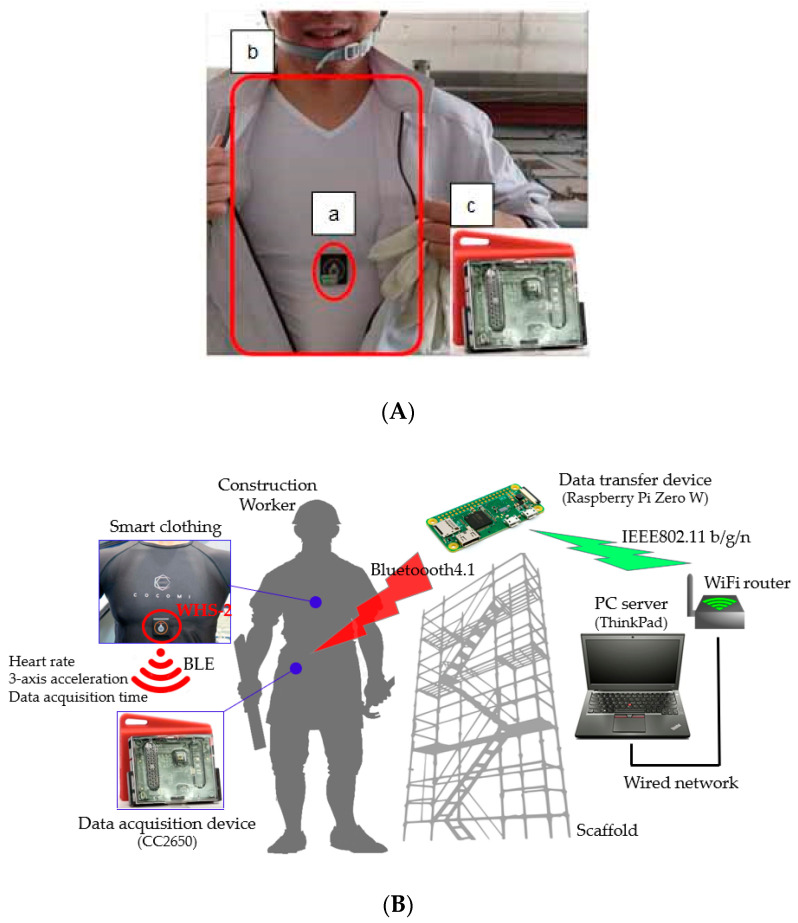
Picture of measurement equipment for physiology in this study. (**A**) a: Heart rate and acceleration sampling sensor (WHS-2), b: smart clothing (COCOMI), c: data acquisition device (CC2650); (**B**) System configuration for physiology in this study.

**Figure 2 sensors-20-03786-f002:**
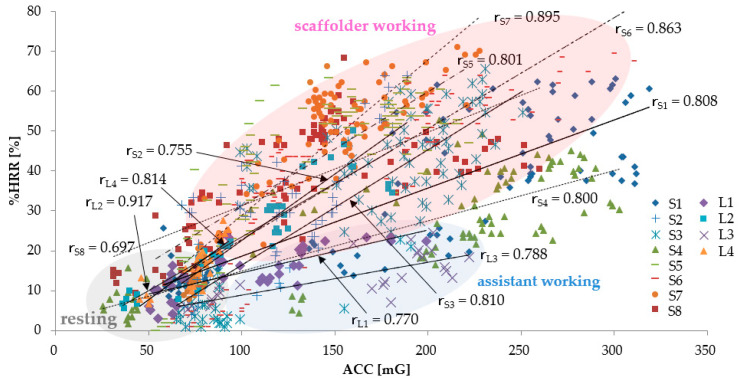
Relationship between acceleration (ACC) and heart rate reserve (%HRR) (Scaffolder_S1–S8_, Labor_L1–L4_). **r**_S1–S8_ denotes the correlation coefficient of the relationship between ACC and %HRR for S1–S8, and **r**_L1–L4_ is the correlation coefficient of the relationship between ACC and %HRR for L1–L4.

**Figure 3 sensors-20-03786-f003:**
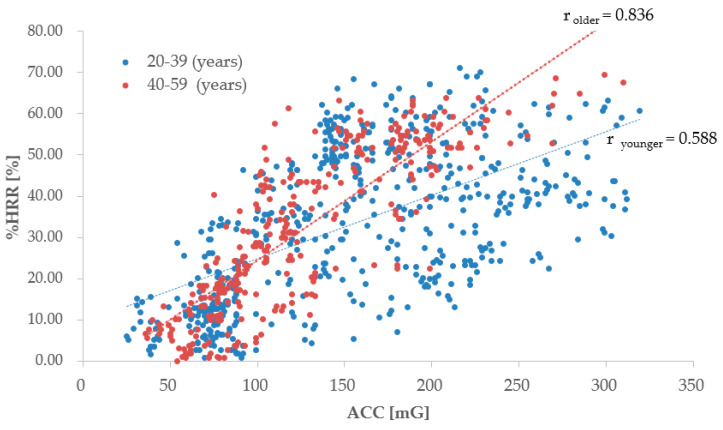
Relationship between ACC and %HRR (workers’ age level are 20–39 years-old and 40–59 years-old. **r**_20–39_ and **r**_40–59_ are the correlation coefficients of the relationship between ACC and %HRR for 20–39 years-old and 40–59 years-old, respectively.

**Figure 4 sensors-20-03786-f004:**
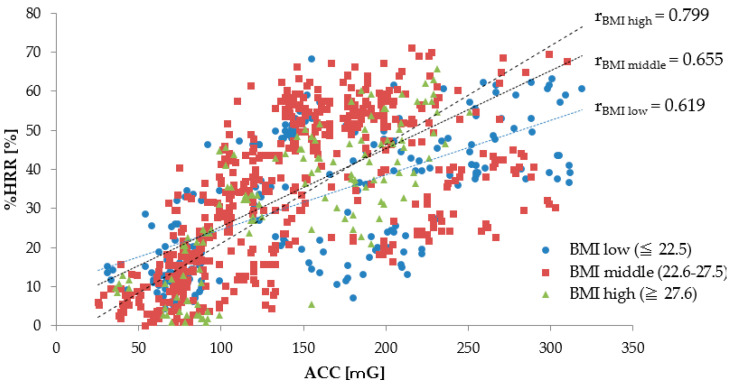
Relationship between ACC and %HRR (Workers’ BMI levels are ≤22.5 kg/m^2^, 22.5–27.5 kg/m^2^, and ≥27.6 kg/m^2^). r_low,_ r_middle_, and r_high_ are the correlation coefficients of the relationship between ACC and %HRR, for ≤22.5 kg/m^2^, 22.6–27.5 kg/m^2^, and ≥27.6 kg/m^2^.

**Figure 5 sensors-20-03786-f005:**
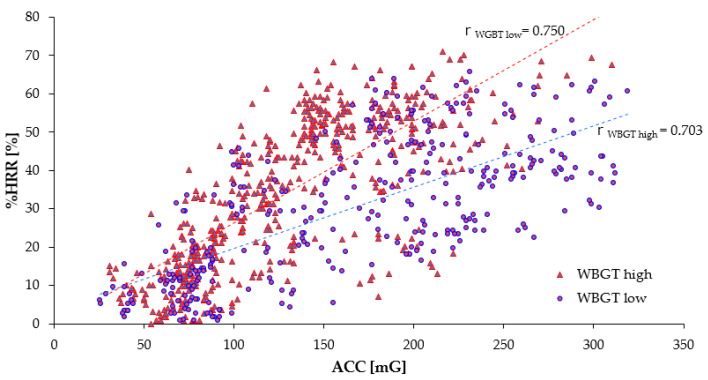
Relationship between ACC and %HRR (WBGT levels are ≤25.8 °C and ≥25.9 °C). **r**_WBGT high_, **r**_WBGT low_ are the correlation coefficients of the relationships between ACC and %HRR for ≤25.8 °C, ≥25.9 °C.

**Table 1 sensors-20-03786-t001:** List of measurement equipment used for measuring meteorology and physiology.

Measurements	Equipment Model (Name of the Manufacturer)	Accuracy	Resolution	Interval	Note
Working environment					
Air temperature	AD-5696(A&D Co., Ltd.)	±1 °C	0.1 °C	10 min.	Thermister
Relative humidity		±5% Rh	0.1% Rh	10 min.	Capacitance
WBGT		--	0.1 °C	10 min.	--
Physical workload					
ECG(Smart clothing)	COCOMI ^*1^ (TOYOBO Co., Ltd.)	1 Ω/sq^*2^	0.3 mm ^*3^	--	Stretchable conductive film
Heart rate sensor	WHS-2 ^*4^ (Union Tool Co., Ltd.)	--	1 kHz ^*5^	Per beat	Analysis of R-R interval
3-axis acceleration		--	31.25 Hz ^*5^	Per beat	Capacitive sense
Infrastructure					
Data acquisition time	CC2650 ^*6^ and ThinkPad (Texas Instruments and Lenovo co., Ltd.)	--	1 ms	Per beat	Synchronized time with server
Data transfer	Raspberry Pi Zero W (Raspberry Pi Foundation)	--	--	--	IEEE802.11 b/g/n (Wireless LAN) Bluetooth 4.1

Note: ^*1^
Figure 1b; ^*2^ sheet resistance of smart clothing; ^*3^ Approximate thickness; ^*4^
Figure 1a; ^*5^ sampling in frequency; ^*6^
Figure 1c.

**Table 2 sensors-20-03786-t002:** Measurement parameters and measurement methods.

Measurement Parameter [Unit]	Method	Unit
BMI	weight/(height)^2^	kg/m^2^
HR_working_	average heart rate in 5 min during working hours	bpm
HR_resting_	average heart rate in 5 min during the rest hours	bpm
HR_max_	208 − 0.7 × age	bpm
%HRR	$\%HRR=\frac{HR\;working-HR\;resting}{HR\;max.-HR\;resting}\times 100$	%
ACC	$\sqrt{{\left(A_{X_{n}}-A_{X_{n-1}}\right)}^{2}+{\left(A_{Y_{n}}-A_{Y_{n-1}}\right)}^{2}+{\left(A_{Z_{n}}-A_{Z_{n-1}}\right)}^{2}}$	mG

**Table 3 sensors-20-03786-t003:** Description of the collected data on the subjects.

ID#	Data Collection Dates	Age (Years)	Main Job Task	Height (cm)	Weight (kg)	Duration of Data Collection (min)	Scheduled Resting (min)	Number of Data * (Sets)
S1	June-29-2018	20	Scaffolder	159.0	57.0	450	90	90
S2	June-29-2018	39	Scaffolder	179.0	74.0	300	90	60
S3	June-29-2018	32	Scaffolder	177.0	93.0	450	90	90
S4	June-29-2018	25	Scaffolder	182.0	82.0	450	90	90
S5	Nov-15-2018	41	Scaffolder	176.0	70.0	510	90	102
S6	Nov-15-2018 Nov-15-2018	40	Scaffolder	176.0	75.0	510	90	102
S7		36	Scaffolder	170.0	68.0	510	90	102
S8	Nov-15-2018	22	Scaffolder	165.0	55.0	510	90	102
L1	May-25-2018	43	Worker	168.0	70.0	210	60	42
L2	May-25-2018	50	Worker	174.5	87.5	210	60	42
L3	May-25-2018	27	Worker	170.5	62.5	150	60	30
L4	Nov-15-2018	59	Worker	169.0	76.0	150	60	30

Note: * The time interval of one data is 5 min.

**Table 4 sensors-20-03786-t004:** Descriptive statistics and correlation matrix for workers’ risk.

Variables	ACC	BMI	AGE	WBGT	VIF
ACC	1.00				1.17
BMI	0.021	1.00			1.48
AGE	−0.333 ***	0.415 ***	1.00		1.43
WBGT	−0.067 *	−0.387 ***	−0.048 *	1.00	1.20

Note: *** indicates *p* < 0.001 and * indicates *p* < 0.05.

**Table 5 sensors-20-03786-t005:** Overview of subjects’ physical workload measurement.

ID#	Estimated HR_max_ (bpm)	Estimated HR_resting_ (bpm)	HR_working_ Ave. ± SD (bpm)	%HRR Ave. ± SD (%)	ACC Ave. ± SD (mG)	r_ACC-%HRR_
S1–S8	185.8 ± 5.6	75.8 ± 2.5	115.0 ± 20.2	35.8 ± 18.5	152.1 ± 67.0	-
S1 ^*2^	194.0	77	117.9 ± 21.4	35.0 ± 18.3	195.6 ± 86.3	0.808
S2 ^*3^	180.7	79	106.6 ± 16.0	27.2 ± 15.8	113.9 ± 35.7	0.755
S3 ^*2^	185.6	76	112.7 ± 21.9	33.4 ± 20.0	158.8 ± 55.7	0.810
S4	190.5	75	105.0 ± 15.8	26.0 ± 13.7	188.0 ± 86.9	0.800
S5 ^*1^	179.3	75	119.1 ± 17.6	42.3 ± 16.9	138.6 ± 47.1	0.801
S6 ^*2^	180.0	72	111.5 ± 22.4	36.5 ± 20.7	149.2 ± 64.4	0.863
S7 ^*1^	182.8	80	125.7 ± 19.4	44.5 ± 18.9	139.9 ± 43.4	0.895
S8 ^*2^	192.6	74	117.2 ± 17.4	36.4 ± 14.7	127.4 ± 55.5	0.697
L1–L4	176.5 ± 7.6	77.0 ± 2.0	95.0 ± 10.7	18.4 ± 10.2	106.7 ± 47.4	-
L1	174.4	76	90.4 ± 5.4	14.2 ± 5.3	99.1 ± 39.5	0.770
L2	173.0	80	106.4 ± 11.2	28.3 ± 11.9	108.2 ± 41.2	0.917
L3	189.1	75	89.8 ± 6.9	13.0 ± 6.1	147.8 ± 57.6	0.788
L4	166.7	76	90.4 ± 5.5	15.9 ± 6.1	74.2 ± 14.8	0.814
Total Ave. ± SD	182.4 ± 8.4	76.3 ± 2.4	111.7 ± 20.4	32.9 ± 18.6	144.7 ± 60.3	-

Note: ^*1^ Heart rate ≥40% HRR all day; ^*2^ ≥30% HRR frequently occurs; ^*3^ working foreman.

**Table 6 sensors-20-03786-t006:** Overview of subjects’ age and relationship between ACC and %HRR.

Age Groups (Years)	Number of Data (Sets)	AGE Average ± SD (Years)	ACC Average ± SD (mG)	%HRR Average ± SD (bpm)	r_ACC-%HRR_
AGE_younger_	562	28.7 ± 7.2	154.7 ± 69.8	33.3 ± 18.3	0.588
AGE_older_	310	46.6 ± 8.0	127.1 ± 55.7	32.3 ± 19.1	0.836
*p*-value_younger-older_ *			<0.001	0.049	-

Note: AGE_younger_ and AGE_older_ denote the group of 20–39 years-old and 30–39 years-old workers, respectively. *: The *p*-value is obtained from multiple comparisons between AGE_younger_ and AGE_older_, by using the Mann–Whitney U test.

**Table 7 sensors-20-03786-t007:** Overview of subjects’ BMI and relationship between ACC and %HRR.

BMI Groups (kg/m^2^)	Number of Data (Sets)	BMI Average ± SD (kg/m^2^)	%HRR Average ± SD (bpm)	ACC Average ± SD (mG)	r_ACC-%HRR_
BMI_low_	222	21.3 ± 1.07	32.6 ± 17.3	157.8 ± 76.6	0.610
BMI_middle_	528	23.9 ± 1.02	33.6 ± 19.3	139.7 ± 63.3	0.767
BMI_high_	132	29.4 ± 0.46	31.8 ± 17.9	142.8 ± 56.6	0.810
*p*-value *_low-middle_			n.s.	0.011	
*p*-value *_middle-high_			n.s.	0.316	
*p*-value *_low-high_			n.s.	0.334	

Note: BMI_low_, BMI_middle_, and BMI_high_ denote ≤22.5 kg/m^2^, 22.6–27.5 kg/m^2^, and ≥27.6 kg/m^2^, respectively. *: The *p*-value is obtained from multiple comparisons among BMI groups by using the Kruskal–Wallis test.; n.s.: non-significant.

**Table 8 sensors-20-03786-t008:** Overview of wet bulb globe temperature (WBGT) and the relationship between ACC and %HRR.

WBGT Groups (°C)	Number of Data (Sets)	WBGT Average ± SD (°C)	%HRR Average ± SD (bpm)	ACC Average ± SD (mG) (mG)	r_ACC-%HRR_
WBGT low	552	16.3 ± 2.75	30.7 ± 17.6	168.6 ± 77.4	0.703
WBGT high	330	27.6 ± 0.21	34.3 ± 19.1	130.5 ± 53.9	0.750
*p*-value * low-high			0.005	<0.001	

Note: WBGT_low_ and WBGT_high_ are <25.9 °C and ≥25.9 °C on WBGT, respectively. *: The *p*-value is obtained from multiple comparisons between WBGT_low_ and WBGT_high_ by using the Mann–Whitney U test.

**Table 9 sensors-20-03786-t009:** Estimation by logistic regression model.

Model	Independent Variables	Coefficient	Standard Error	Wald χ^2^	*p*-Value	Odds Ratio	95% CI for Odds
Model 1	Constant	−25.6	3.40	61.3	<0.001	0.000	–
	ACC	0.041	0.003	183	<0.001	1.042	1.036–1.048
	AGE	0.074	0.020	14.0	<0.001	1.077	1.036–1.120
	BMI	−0.035	0.058	0.361	0.548	0.966	0.862–1.082
	WBGT	0.705	0.095	54.6	<0.001	2.023	1.678–2.438
Model 2	Constant	−28.1	2.35	142	<0.001	0.000	–
	ACC	0.041	0.003	181	<0.001	1.042	1.036–1.048
	AGE	0.066	0.014	23.1	<0.001	1.068	1.040–1.097
	WBGT	0.742	0.074	101	<0.001	2.100	1.817–2.427

**Table 10 sensors-20-03786-t010:** Overview of subjects’ physical demand measurement.

Model (Independent Variables)	Predicted Risk			Percentage	GoF		
	Observed	0	1	(%)	AIC	Cox–Snell R^2^	Nagelkerke R^2^
Model 1 (ACC, AGE, BMI, WBGT)	Risk 0	487	53	90.2	57.5	0.435	0.590
	1	45	297	86.8			
	Overall			88.5			
Model 2 (ACC, AGE, WBGT)	Risk 0	486	54	90.0	59.5	0.434	0.589
	1	40	302	88.3			
	Overall			89.2			

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
