# Peer review of "Practical Judgment of Workload Based on Physical Activity, Work Conditions, and Worker’s Age in Construction Site"

_sensors, 2020, doi:10.3390/s20133786_

Round 1

Reviewer 1 Report

Dear Authors,

I read your re-submission of the manuscript "Practical Judgment of Workload Based on Physical Activity, Work Conditions, and Worker’s Age" and your replies to my observations. Congratulation because now the article has been significantly improved and I believe it can be ready for publication.

Best regards,

the Reviewer 

Author Response

June 22, 2020

Dear Reviewer,

First of all, we would like to express to you our gratitude. Your comments and suggestions are invaluable for us to improve our paper. We express our sincere thanks to reviewer for your advices.

Please see the attachment in detail.

We highly appreciate your cooperation.

Best Regards,

Nobuki Hashiguchi (N. H.)

Graduate School of Technology Management, Ritsumeikan University

2-150, Iwakura-cho, Ibaraki, Osaka, 567-8570, Japan

Tel.: +81-72-665-2100

Kota Kodama

Graduate School of Technology Management, Ritsumeikan University

Osaka, Japan

Email: [email protected] (K.K.)

Tel.: +82+72-665-2448

Reviewer 2 Report

The present study provides a physiological, environmental, and motion sensor-based analysis and prediction model for workload in construction workers. The manuscript is well-written and the study details have been presented clearly and scientifically sound. I think it deserves to be published unaltered. 

Author Response

June 22, 2020

Dear Reviewer,

First of all, we’d like to express to you our gratitude. Your comments and suggestions are invaluable for us to improve our paper. We express our sincere thanks to reviewer for your advices.

Please see the attachment in detail.

We highly appreciate your cooperation.

Best Regards,

Nobuki Hashiguchi

Graduate School of Technology Management, Ritsumeikan University

2-150, Iwakura-cho, Ibaraki, Osaka, 567-8570, Japan

Tel. +81-72-665-2100

Kota Kodama

Graduate School of Technology Management, Ritsumeikan University

Osaka, Japan

Email: [email protected] (K.K.)

Tel.: +82+72-665-2448

This manuscript is a resubmission of an earlier submission. The following is a list of the peer review reports and author responses from that submission.

Round 1

Reviewer 1 Report

The authors conducted a study on construction site workers to estimate workload risk utilizing physical activity, the wet bulb globe temperature (WBGT) of the workplace,  workers' age, and BMI. They found that all the factors except for BMI can help assess the workload risk. This study contributes to the field by using data collected from a light-weight setting (wearable device) for real-time monitoring to minimize the workers' health damage. However, to be published, minor revisions are needed as below.

- Abstract has the work real-time method. It is misleading that I first expected that the data analysis and alert was given real-time.
- In the abstract smart clothing can be just written wearable device
- The title format for the Tables are different. Some have all the words capatalized and some are not. Some are bold and some are not.
- Figure 1. It would help if they explained what a, b, c is in the caption.
- there are commas that have a different format.
- Table 7. n.s. -> even thought they are not significant, we usually report the value and mark the ones that are significant with *.

Line by line
124: table -> Table
326 - comma oddly shown
418: pace -> place

Reviewer 2 Report

The subject matter of the article "Real-time judgment of workload based on physical activity, work conditions, and worker’s age" is important and relevant.

Extending the conclusions – what are the additional topics that required for review and how the article promotes science. English language and style are fine/minor spell check required.

Reviewer 3 Report

Dear Authors,

I am sorry but I have to say that your paper is currently not ready to be cosidered for possible publication in IJERPH.

I suggest to you to carefully re-write your manuscript following STROBE guidelines for reporting observational studies, and also to strictly follow the indications on how to write the different sections of a scientific paper. 

I noticed that your introduction is too long and not focused on clearly defining and setting the adequate scientific background on the topic of interest.

The objectives of the study are not clearly identifiable. material and methods section is confusing, it does not follow an adequate order of presentation, an many results are inappropriately presented in this section.

The results are misleading, with too much tables and figures: again it is not clear what are the main goals and messages of your study.

The discussion, on the other hand, is too short, with no description of the study limits, while conclusions, again, are too long, let me thinking that the finale message of the study is not clear.

References are quite outdates: approximately only ten on 70 references from 2017 and 2018. It seems to me no references at all from 2019 and 2020: an effort is needed to insert your study in recent scientific evidence.

I will be happy to revise again a new version of your manuscript in the future.

I wish you a good job.

Best regards,

the Reviewer